# All You Need Is A Reference: Cross-modality Referring Segmentation for Abdominal MRI

## Abstract

Multi-modality MRI scans can provide comprehensive diagnoses of abdominal disease but this also introduces new segmentation burdens to quantitative image analysis. In this work, we introduce an image-based referring segmentation task where users only need to draw simple scribbles on one modality (reference modality), to guide the segmentation of multiple unseen target modalities. To benchmark the multi-modality segmentation task, we annotate a new dataset with 3,277 organs from 534 MRI images, covering five commonly used MRI modalities. Furthermore, we present a referring segmentation model, CrossMR, to simultaneously segment multiple target modalities based on scribbles on reference modality only. Experiments demonstrate that our method can achieve comparable performance to the state of the art on one in-distribution reference modality and significantly better generalization ability on four out-of-distribution target modalities. The new dataset, code, and trained model weights will be publicly available at `https://ref-seg-mr.github.io/`.

## 1 Introduction

Magnetic Resonance Imaging (MRI) has been a widely used medical imaging technology, offering a non-invasive method for visualizing internal structures of the human body. In abdominal imaging, MRI stands out for its exceptional soft tissue contrast and the ability to produce images in any anatomical plane without radiation. Moreover, the development of multi-modality MRI techniques, such as T1-weighted (T1w) MRI, T2-weighted (T2w) MRI, and Diffusion-Weighted Imaging (DWI), provides a more comprehensive evaluation of diseases and significantly enhances diagnostic accuracy. These features make multi-modality MRI particularly suitable for the clinical examination of abdominal diseases Mulé et al. (2020).

Quantitative imaging biomarkers measured with MRI are emerging as important clinical tools in the evaluation and management of abdominal diseases, such as liver fibrosis and fatty liver Guglielmo et al. (2023); Xia et al. (2023). However, producing these image biomarkers usually requires segmentation of the target organs. For example, the measurements of fat fraction and iron concentration rely on the liver segmentation mask Martí-Aguado et al. (2022). In clinical practice, patients usually undergo multiple MRI modalities. Manually segmenting the target organs in each modality would be a labor-intensive and time-consuming task.

This work aims to formulate and address the question: **how to efficiently segment multi-modality MRI leveraging annotations of a single reference modality?** We first annotate a multi-modality MRI dataset consisting of five modalities, where T1w modality is used as the reference modality with annotations and the other four modalities, T2w, DWI, In-phase, and Opposed-phase MRI, do not have annotations, as shown in Fig. 1 (a). Our goal is to leverage only the annotations of the reference T1w modality for model training to segment all five modalities during inference.

Compared to traditional unsupervised cross-modality domain adaptation tasks, our task design has two unique features. First, the goal is to segment both the source-domain (i.e. T1w) as well as multiple target domains (i.e. T2w, DWI, In-phase, and Opposed-phase) simultaneously with only one suit of model parameters while traditional tasks mainly focus on one target modality. Second, we frame the cross-modality segmentation as a referring segmentation task, where the weak annotation on the reference modality guides the segmentation of the target modalities. As this task involves

weak annotation of a single modality, such as widely used T1w modality, the annotation burden is significantly lower.

In this paper, we introduce the image-based referring segmentation to address the above question, where the annotation from a reference modality guides the segmentation process for a target modality. As illustrated in 1 (b), during inference, clinicians only provide weak annotations, such as simple scribbles, on the target organ (i.e., liver, kidneys, and spleen) of the reference modality (i.e., T1w MRI), to obtain segmentation on all five modalities. We introduce CrossMR, a referring segmentation model that leverages these scribble annotations from the reference modality to accurately segment all five modalities. We summarize the main contributions as follows:

- **Novel task.** This work introduces the first referring segmentation task for organ segmentation in multi-modality MRI. This also poses a unique segmentation challenge where models need to segment multiple target images based on weak annotations in one reference image.

- **Annotated multi-modality MRI dataset.** We create a labeled dataset that contains 534 cases with 3,277 annotated segmentation masks, covering four abdominal organs across five imaging modalities. The dataset will be made publicly available to the community for research purposes, facilitating further exploration of the referring segmentation in medical imaging.

- **Generalizable algorithm: CrossMR.** We present a referring segmentation model, which enables users to input scribble prompts on a reference image to simultaneously deliver segmentation results of the multiple target images.

## 2 RELATED WORK

**Cross-modality Domain Adaptation Method** Domain adaptation is a common approach to cross-modality segmentation task. CrossMoDA challenge hosts a benchmark of cross-modality unsupervised domain segmentation method where the task is to leverage the labeled source domain T1w data and the unlabeled target domain T2w data to build a segmentation model for the target domain. The solutions employ unpaired image translation such as cycleGAN Zhu et al. (2017) and QS-Attn Hu et al. (2022), followed by segmentation model training with the synthesized images and self-training Liu et al. (2023). The goal is to build a specialized segmentation model focused on the performance of a single target modality and usually involve a multi-step approach specifically tailored for the target domain modality. compared to existing cross-modality domain adaptation methods Guan & Liu (2021), our proposed referring segmentation task generalizes to more than two modalities where the aim is to build an end-to-end model that has competitive performance across all MR modalities, emphasizing the importance of efficiently handling the inherent heterogeneity of medical images.

**Few Shot and One Shot Segmentation** To improve model generalizability to unseen modalities or tasks without supervised fine-tuning, few-shot or one-shot approaches have been explored (Butoi et al., 2023; Wu & Xu, 2024). These methods input a query image alongside one or a few image-label pairs, using this support information to guide accurate segmentation of the query image. Recently, PerSAM, a one-shot approach capable of zero-shot segmentation with or without fine-tuning on a single reference sample, has been proposed (Zhang et al., 2023). Built on SAM (Kirillov et al., 2023), PerSAM customizes SAM for personalized object segmentation using target-specific attention and semantic prompting without additional training. PerSAM-F further improves segmentation by introducing a quick, scale-aware fine-tuning process, adjusting only two parameters in 10 seconds for more precise results. However, while these methods efficiently utilize the support set's annotation, they often require fully annotated data, and some approaches, like Butoi et al. (2023), rely on larger support sets of size 64 to achieve better performance.

**Prompting in Foundation Models** An emerging approach to generalize the model capabilities is to utilize foundation models, built from large-scale datasets. Some foundational segmentation models employ prompting to guide the model to adapt to the task during inference. MedSAM Ma et al. (2024), trained on diverse medical datasets, uses an input bounding box to prompt the model with the region of interest. As scribble prompts are suited for segmentation of complex Luo et al. (2022), scribble prompt models trained on a diverse set of medical imaging are also introduced Marinov et al. (2024); Wong et al. (2023). These models achieve state-of-the-art performance on various

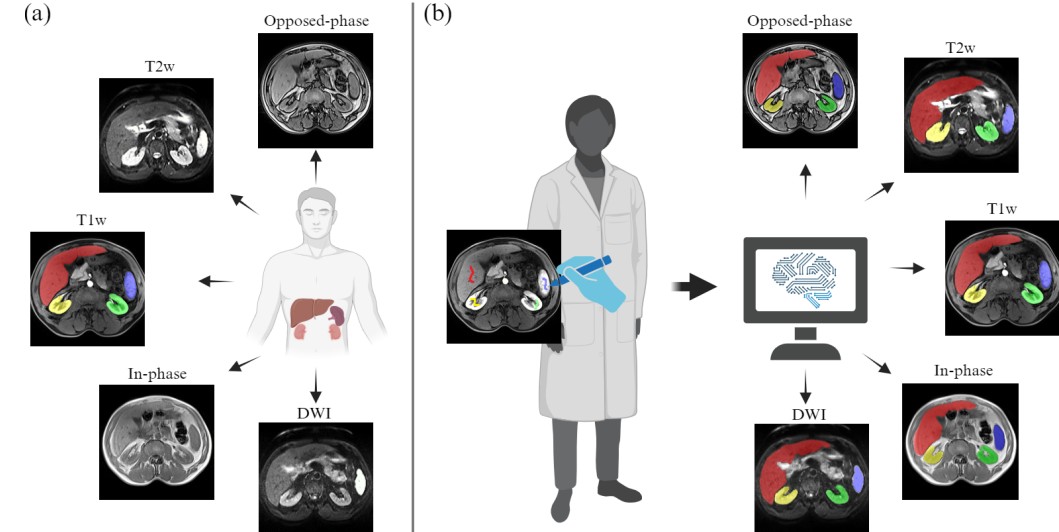

Figure 1: **Task definition of referring segmentation in multi-modality MRI images.** (a) Patients undergo five modalities, including one reference modality T1w and four target modalities: T2w, DWI, In-phase, and Opposed-phase. The task is to generate segmentation masks for all the target modalities based on one reference modality. (b) Clinical application scenarios: Users first draw scribbles on the reference modality. Then, the model simultaneously outputs segmentation masks for all five MR modalities corresponding to the scribbled organs.

medical imaging datasets. However, prompts are required for each target in each image, which can be inefficient, especially in scenarios with multiple targets or when processing large volumes of data. Our approach uses one prompt across all target modalities, which significantly reduces the annotation burden.

**Referring segmentation** The key idea of referring segmentation is to leverage the image feature corresponding to the local scribbled region of the reference image to gain information on the organ of interest on the target image. It focuses on the efficient use of annotations as the scribble annotation is only necessary on the reference image. Using weak annotation of one reference image, referring segmentation aims to simultaneously generate masks for multiple target images. Referring segmentation has emerged as a popular task in nature image segmentation Zou et al. (2024); Li et al. (2023), but it has not been well formulated and studied in the medical imaging field. We adapt referring segmentation in cross-modality segmentation and present a referring segmentation model, CrossMR, which is specifically tailored to reduce the workload on clinicians by simplifying the annotation process while ensuring high-quality segmentation outcomes across different imaging modalities.

## 3 METHOD

We choose the Segment Everything Everywhere Model (SEEM) Zou et al. (2024) as the base model because it supports general prompts and has better flexibility than the Segment Anything Model (SAM) Kirillov et al. (2023). Then, we build CrossMR with two task-specific modifications: paired data augmentation and the organ-specific bipartite matching.

### 3.1 ARCHITECTURE

Fig. 2 shows the overall architecture of CrossMR, consisting of the image encoder, visual sampler, and decoder. The input contains paired images, either from the paired data augmentation during training or from paired target-reference images during inference. The image encoder used is the FocalNet Yang et al. (2022), which accepts both reference and target images to generate image features $\boldsymbol{F}^{ref} \in \mathcal{R}^{C \times H \times W}$ and $\boldsymbol{F}^{target} \in \mathcal{R}^{C \times H \times W}$, respectively. Then, the visual sampler aims

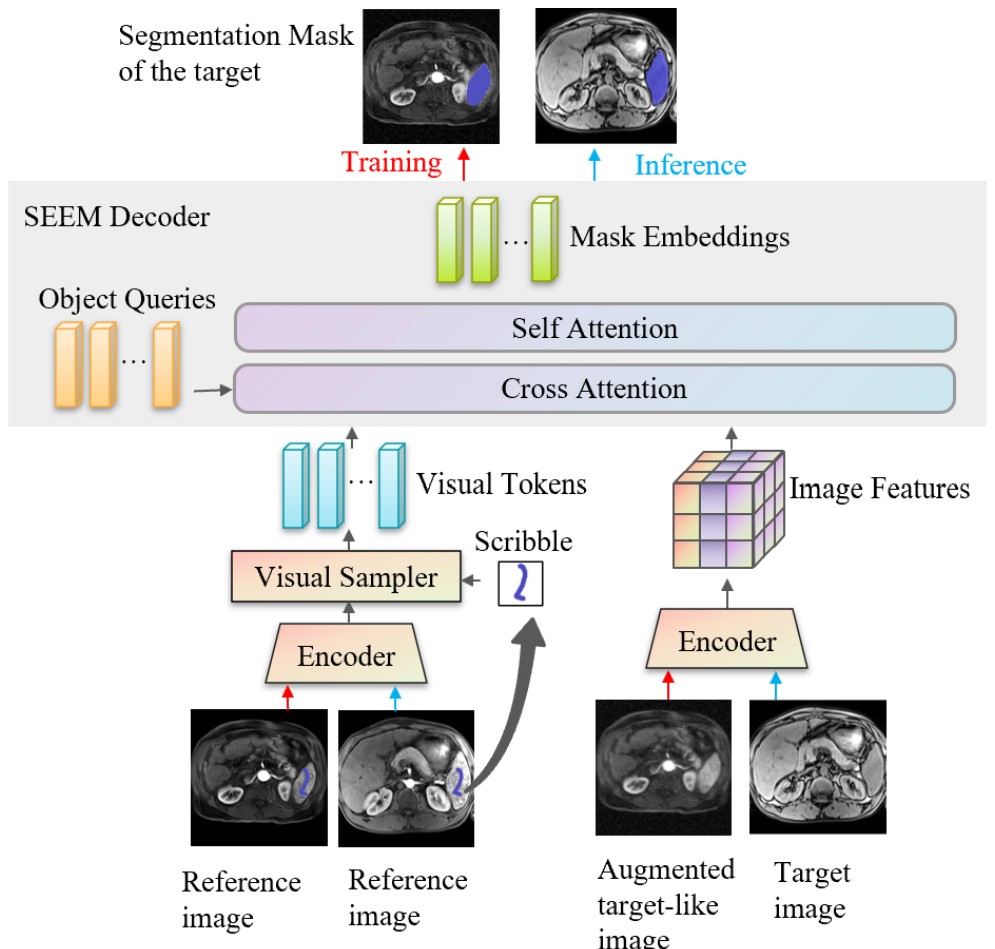

Figure 2: **CrossMR Architecture.** During training, indicated by red arrows, the reference image and the augmented target-like image are each input to the model to output segmentation of the target-like image. During inference, indicated by blue arrows, the paired reference and the target image are input to the encoder to output the target segmentation. The scribble prompt drawn on the reference image is input to the visual sampler to extract visual tokens. Learnable object queries are duplicated for each organ and concatenated with visual tokens. The combined queries and visual tokens cross-attend to the reference image features and self-attend within themselves. The refined object queries are linearly mapped to form mask embeddings which are used to output the segmentation mask.

to extract the visual tokens, which uses an interpolation function to sample the features from the scribbled region.

In the decoder, learnable object queries are employed to allow the model to learn the semantics of the objects. The object queries are duplicated for each organ to form spatial queries, which are separately used to gain information about each organ. The spatial queries $\boldsymbol{Q}$ interact with the target image features by cross-attention to $\boldsymbol{F}^{target}$. Next, the visual tokens for each organ are concatenated with the spatial queries of the organ and self-attention is performed within the combined queries, aligning both the target and reference features. The output queries from the spatial queries are linearly mapped to mask embeddings $\boldsymbol{M} \in \mathcal{R}^{N,C}$. The dot product along the channel dimension between the mask embedding and the image features outputs the binary mask for each object. For the i-th query, the resulting binary mask $\boldsymbol{B}_{i,h,w} \in \mathcal{R}^{N,H,W}$ is computed by:

$$\boldsymbol{B}_{i,h,w} = \sum_{c=1}^{C} \boldsymbol{M}_{i,c} \cdot \boldsymbol{F}^{target}_{c,h,w}, \quad \text{for } i = 1, \dots, N$$

where $N$ is the number of object queries. The loss function follows the extension of the mask classification Cheng et al. (2021) loss, which is a weighted sum of the binary cross-entropy loss and the dice-entropy loss.

## 3.2 Paired Data Augmentation

To incorporate image features of diverse contrast and intensity that closely aligns with the features of the target modalities during training, paired data augmentation is used. For each reference modality input, two independent sets of data augmentations are used to produce two images. The first augmentation set traslates the T1w reference image to an image that resembles one of the five modalities. The second augmentation generates a reference T1w image with no augmentation or a slight change in contrast or intensity.

Through paired data augmentation, CrossMR learns to integrate reference image features with scribble annotations by allowing the queries to attend to both the scribble and the target image features. This helps to incorporate and merge image features of the paired images and aligns the augmented image representations without the need of using separate generative model for image synthesis. In the following sections, we refer to both the target image used during inference and the augmented, target-like images generated during training simply as the target image.

## 3.3 Organ-specific Bipartite Matching

To build a one-to-one matching between the ground truths and the binary predictions, bipartite matching is commonly used in mask classification models Cheng et al. (2021); Zou et al. (2023). As the predictions are made from the spatial queries, it is crucial that the relevant spatial queries for each organ are matched with the ground truth organ. Therefore, we introduce an organ-specific bipartite matching, where only queries that self-attend to the visual tokens of the corresponding organ are considered to be matched.

For the binary target $\boldsymbol{T} \in \mathcal{R}^{K,H,W}$ with $K$ organs and the predicted masks $\boldsymbol{Y} \in \mathcal{R}^{N \times K,H,W}$, the bipartite matching cost matrix $\boldsymbol{C}^{match}$ of size $(N \times K, K)$, where $(i, j)$ is the cost of i-th spatial query and j-th object, is computed for each $K$ target mask and each $N \times K$ spatial query as the weighted average of the binary cross-entropy loss and the dice loss:

$$\boldsymbol{C}^{match}(\boldsymbol{Y}, \boldsymbol{T}) = \beta' \boldsymbol{L}_{BCE}(\boldsymbol{Y}, \boldsymbol{T}) + \gamma' \boldsymbol{L}_{DSC}(\boldsymbol{Y}, \boldsymbol{T})$$

To ensure that the linear assignment problem Virtanen et al. (2020); Crouse (2016) does not match the queries corresponding to queries that self-attended to a different organ, we set the corresponding values to infinity to disable matching.

$$\boldsymbol{C}_{i,j}^{match}(\boldsymbol{Y}, \boldsymbol{T}) = \infty, \quad \forall i \in [0, K), i \notin [N \times j, \ N \times (j+1))$$

In addition, to build object queries that can take in object information through attention layers efficiently, we do not use the random sampling of the object queries used in SEEM but instead use all object queries for each training iteration.

## 3.4 Automatic Scribble Generation

We follow the implementation of automatic scribble generation from SEEM, based on the method proposed in Yu et al. (2019). The method aims to generate flexible and controllable scribbles similar to real use cases on-the-fly. From a uniformly sampled start point, it randomly selects an angle and a length to draw, using a variable brush width within a specified range. After drawing a line of the selected length in the chosen direction, it samples the angle and length again to continue drawing from the endpoint. To ensure smooth transitions between lines, a circle with a randomly selected size within the brush width range is drawn between the two lines. Finally, small random displacements are applied to a subset of the control points, creating a more organic, free-form appearance.

# 4 EXPERIMENTS

## 4.1 DATASET

Duke Liver dataset Zhu et al. (2020); Macdonald et al. (2023) was used in this work, which is a multi-planar, multi-phase, and multi-contrast dataset with 2,146 abdominal MRI scans from 105 subjects. We selected the axis imaging modalities because they are usually used for extracting quantitative imaging biomarkers Guo et al. (2023). We also excluded the modalities with less than 50 scans, which is not enough to draw statistically significant results. Finally, we obtained five modalities: T1w, T2w, DWI, In-phase, and Opposed-phase, including 534 cases in total. We used the T1w MRI as the reference modality because of its popularity in clinical practice and the remaining four modalities as the target modalities.

The original dataset only provided limited liver masks for T1w images. However, the segmentation of the other organs in multi-modality MRI also plays an important role in clinical practice, such as left kidney, right kidney Goel et al. (2022), and spleen Camastra & Ciolina (2024). Therefore, we further produced annotations for all the selected MRI scans, leading to a large-scale labeled dataset with 3,277 organs. The labels were annotated by a junior radiologiest with one year of experience using ITKSNAP Yushkevich et al. (2016) followed by careful examination and revision of a senior radiologist with over ten years of experience in abdominal radiology. The dataset was randomly divided into 80%, 10%, and 10% at the patient level as the training, validation, and testing sets, respectively.

## 4.2 IMPLEMENTATION

During preprocessing, we first clipped the intensity values to the range between the 0.5th and 99.5th percentiles followed by rescaling them to the range of [0, 1]. Then, we resized each image to $256 \times 256$. During training, we applied commonly used data augmentations from MONAI Cardoso et al. (2022), such as random Gaussian noise or gibbs noise, random contrast adjustment, and random scale intensity. The empirical weights of the binary cross-entropy loss $\beta$, and dice loss $\gamma$ were set to 2.5, and 2.5 respectively. For the organ-specific bipartite matching, cross-entropy loss $\beta'$, and dice loss $\gamma'$ are set to 2.0 and 2.0.

We set the number of the object queries $N$ to 11 and the AdamW optimizer Loshchilov & Hutter (2017) with the base learning rate of 1e-3. The model with the lowest validation loss was selected for evaluation. The learning rate of the image encoder was set to 10 times smaller than the base learning rate. The models were trained for a maximum of 50 epochs.

## 4.3 BENCHMARKING

To the best of our knowledge, there is no previous work on specialized image-based referring segmentation models for multi-modality MRI images. As an alternative, we compared CrossMR with three groups of related state-of-the-art segmentation models: the traditional semantic segmentation model, the one-shot segmentation model, and the scribble-prompt segmentation model. We did not consider multi-stage models such as the cross-modality methods that heavily rely on transferring the source domain images into the target domain using image-level alignment. For the semantic segmentation model, we chose nnU-Net Isensee et al. (2021), a popular and high-performing self-configuration framework with U-Net architecture. Specifically, we used the 2D model and trained from scratch with the same data split as the CrossMR. For the one-shot segmentation model, we used PerSAM-F Zhang et al. (2023), a one-shot fine-tuning model. We frame the one-shot model into our task by using the reference-mask pair as the support set to query the target image. PerSAM-F uses the pre-trained SAM-Huge model and fine-tunes the model in test time on the labeled reference. We also compared with another scribble-prompt segmentation model, MedSAMScribble Marinov et al. (2024), the scribble prompt version of the MedSAM Ma et al. (2024). The model is trained with samples of size $256 \times 256$ and the same data augmentation used for the target images in CrossMR is used for consistency. Lastly, we compared with the scribble version of the nnUNet, referred to as nnUNet-Scribble, where the scribbles are input as a second channel of the input in addition to the images. A similar strategy has been demonstrated by the author's team for general medical image segmentation Stock et al. (2024).

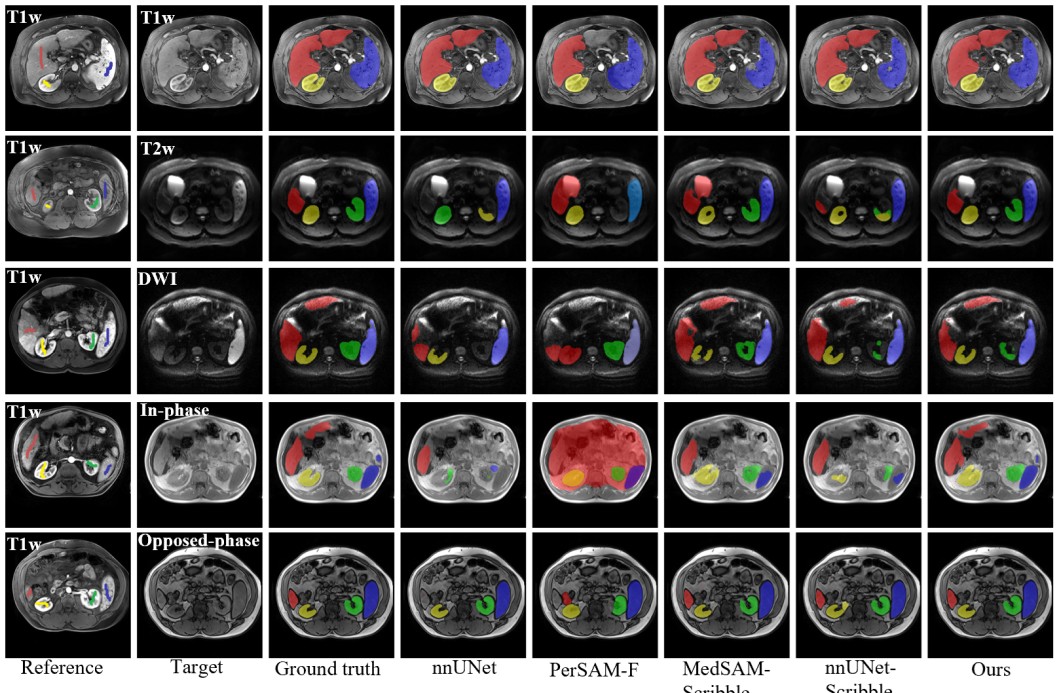

Figure 3: Visualized segmentation examples. The reference image and the scribble prompt (1st column) are used to guide the segmentation of the target images (2nd column). Compared to the other methods, our method achieved competitive performance on the ID modality (1st row), and significantly better segmentation qualities on the OOD modalities (2nd-5th rows).

To evaluate the performance of CrossMR, we randomly grouped the reference T1w images and associated organ scribbles with target images to generate reference-target image pairs. Paired images are chosen from the same subject. The slice pairs were generated by matching slices from the reference volume to slices from the target volume in a one-to-one fashion while maintaining the relative order of the slices. The number of evaluation pairs considered is 4,277 in total with 1,566 for T1w, 457 for T2w, 284 for DWI, 1,969 for In-phase, and 993 for Opposed-phase target images. Since the reference images were from T1w MRI, we refer to the testing T1w images as the in-distribution (ID) modalities and the others as the out-of-distribution (OOD) modalities. The inference was performed by inputting the scribble from only the reference modality to segment the target image.

For the evaluation measures, we followed the recommendations in Metrics Reloaded Maier-Hein et al. (2024) and used the Dice Similarity Coefficient (DSC) and Normalized Surface Dice (NSD), which measure the region similarity and boundary similarity, respectively. The evaluations are performed for each organ and each modality.

### 4.4 QUANTITATIVE AND QUALITATIVE RESULTS

Table 1 shows the quantitative results of CrossMR and the other compared methods. On the ID testing modality T1w, it is expected that nnUNet achieves the best performance because it is a specialized model on this task. Nevertheless, CrossMR obtains high competitive DSC score where the performance difference is marginal across all organs. On the OOD testing modalities, CrossMR significantly outperformed other methods across all the modalities. On average, CrossMR exceeds the second-best performing models by 9.83%, 9.72%, 19.56%, and 5.66% for T2w, DWI, in-phase, and OpposedPhase respectively.

The NSD score, as depicted in Table 3, further highlights the superior performance of our model. The average NSD of our model for each modality consistently surpasses those of the compared methods. This underscores that CrossMR is particularly effective in accurately delineating object

Table 1: **Quantitative segmentation results.** Organ-wise DSC scores (%) of CrossMR and four compared models across one in-distribution modality (T1w) and four out-of-distribution modalities (T2w, DWI, In-phase, and Opposed-phase).The best performance for each modality and organ is highlighted in bold, while the second-best performance is indicated with an underline.

| Modality | Methods | Liver | Right Kidney | Left Kidney | Spleen |
|---|---|---|---|---|---|
| T1w | nnUNet Isensee et al. (2021) | **97.12 ± 6.19** | **98.15 ± 2.60** | **97.40 ± 3.03** | **95.03 ± 12.44** |
| | PerSAM-F Zhang et al. (2023) | 76.36 ± 20.50 | 91.83 ± 15.32 | 92.48 ± 14.38 | 84.66 ± 24.46 |
| | nnUNet-Scribble Stock et al. (2024) | 97.01 ± 9.9 | 97.57 ± 6.54 | 96.98 ± 8.07 | 96.19 ± 10.40 |
| | MedSAMScribble Marinov et al. (2024) | 91.45 ± 5.1 | 91.14 ± 9.3 | 88.65 ± 9.6 | 89.52 ± 15.8 |
| | CrossMR | 96.83 ± 2.58 | 97.56 ± 2.62 | 97.27 ± 4.28 | 94.97 ± 11.70 |
| T2w | nnUNet Isensee et al. (2021) | 72.51 ± 30.74 | 81.43 ± 35.26 | 83.55 ± 31.18 | 86.70 ±23.66 |
| | PerSAM-F Zhang et al. (2023) | 56.67 ± 31.27 | 73.72 ± 35.90 | 65.47 ± 41.24 | 76.13 ± 33.67 |
| | nnUNet-Scribble Stock et al. (2024) | 75.67 ± 12.19 | 93.94 ± 32.19 | 94.06 ± 6.84 | 92.63 ±13.92 |
| | MedSAMScribble Marinov et al. (2024) | 80.42 ± 5.1 | 82.30 ± 9.3 | 87.36 ± 9.6 | 86.60 ± 15.8 |
| | Ours | **85.43 ± 23.84** | **93.96 ± 13.95** | **94.83 ± 9.88** | **93.79 ± 7.38** |
| DWI | nnUNet Isensee et al. (2021) | 71.86 ± 38.85 | 78.76 ± 26.03 | 69.19 ± 30.49 | 93.80 ± 9.01 |
| | PerSAM-F Zhang et al. (2023) | 52.67 ± 25.90 | 58.62 ± 42.82 | 54.07 ± 45.17 | 60.59 ± 43.36 |
| | nnUNet-Scribble Stock et al. (2024) | 81.77 ± 20.72 | 85.14 ± 20.14 | 86.22 ± 21.68 | 93.77 ± 14.31 |
| | MedSAMScribble Marinov et al. (2024) | 80.29 ± 5.1 | 75.56 ± 9.3 | 81.88 ± 9.6 | 87.63 ± 15.8 |
| | Ours | **89.37 ± 9.86** | **88.99 ± 6.26** | **89.98 ± 3.60** | **94.81 ± 8.58** |
| In-phase | nnUNet Isensee et al. (2021) | 54.47 ± 28.65 | 33.97 ± 34.90 | 26.70 ± 33.15 | 28.69 ± 34.17 |
| | PerSAM-F Zhang et al. (2023) | 48.15 ± 28.96 | 33.66 ± 42.23 | 37.84 ± 43.64 | 51.33 ± 35.89 |
| | nnUNet-Scribble Stock et al. (2024) | 57.90 ± 29.47 | 47.96 ± 33.42 | 47.33 ± 28.86 | 40.46± 34.95 |
| | MedSAMScribble Marinov et al. (2024) | 69.58 ± 5.1 | 61.14 ± 9.3 | 74.78 ± 9.6 | 74.75 ± 15.8 |
| | Ours | **84.17 ±15.15** | **80.28 ± 21.73** | **83.88 ± 16.00** | **86.65 ± 16.78** |
| Opposed-phase | nnUNet Isensee et al. (2021) | 84.12 ± 34.53 | 74.44 ± 21.98 | 73.69 ± 36.33 | 79.67 ± 29.32 |
| | PerSAM-F Zhang et al. (2023) | 61.11 ± 30.33 | 53.04 ± 42.63 | 61.66 ± 41.01 | 66.56 ± 35.84 |
| | nnUNet-Scribble Stock et al. (2024) | 86.72 ± 17.49 | 87.94 ± 21.93 | 89.29 ± 17.45 | 87.83 ± 21.24 |
| | MedSAMScribble Marinov et al. (2024) | 87.70 ± 5.1 | 79.34 ± 9.3 | 79.63 ± 9.6 | 84.35 ± 15.8 |
| | Ours | **94.47 ± 5.14** | **90.68 ±7.92** | **89.11 ± 9.91** | **91.68 ± 11.67** |

boundaries, demonstrating its robustness in handling complex anatomical structures with irregular surfaces. Additionally, notable improvements are observed in the in-phase modalities where the average NSD scores are generally lower across all models. Compared to the second-best performing MedSAM-Scribble, CrossMR achieves significantly higher average NSD scores, with increases of 15.98, 23.28, 16.19, and 16.08 points for the liver, right kidney, left kidney, and spleen, respectively. These results emphasize CrossMR's robustness and its ability to deliver substantial performance improvements in challenging OOD modalities.

The superior performance is also evident in the qualitative segmentation outcomes depicted in Fig. 3. We observe that the performance of our model on the ID modality is comparable to the top performing nnUNet, segmentation masks closely matching the ground truth. For the OOD modalities, our model's segmentation masks exhibit clearer boundaries and higher overlaps with the ground truth. In contrast, nnUNet displays misclassification in T1w, with the right kidney being misclassified as the left kidney, and the left kidney as the right. The PerSAM-F model results show over-segmentation, particularly in the T2w and in-phase liver segmentation. Additionally, it misclassifies the left kidney as the spleen in T2w and the right kidney as the liver in DWI, while failing to localize the liver in Opposed-phase. MedSAM-Scribble tends to produce less accurate boundaries and over-segment the liver in T2w and DWI. Lastly, nnUNet-Scribble shows misclassification of the left kidney as the right kidney in T2w and under-segmentation of the kidneys in in-phase.

## 4.5 ABLATION STUDY

We considered five different settings for CrossMR. First, we evaluated the impact of attending to the visual tokens of the scribble reference by removing them from the attention mechanism. Specifically, in the self-attention layer for spatial queries, the visual tokens were excluded, so each organ's spatial queries only attended to themselves. This variant allowed us to assess the effect of the reference scribble on overall performance. Second, we evaluated the model without the organ-specific bipartite matching algorithm to assess its advantages compared to the original bipartite matching algorithm. Third, to assess whether a single augmented reference image can be used to capture the diverse representations of the target images, we evaluated the model without paired augmentation. Fourth, to evaluate whether using only the visual tokens coming from the same source image is a suitable approach to constructing visual tokens, we constructed a version that self-attends to visual tokens across all images in a batch for each organ, drawing inspiration from DINOv Li et al. (2024). For

Table 2: **Ablation study: the performance of CrossMR with different settings**. SEEM (Baseline): SEEM model  Zou et al. (2024), w/o Visual Tokens: without self-attention to visual tokens, w/o Organ Match: without organ-specific bipartite matching, w/o Paired Aug.: without the paired augmentation, Visual Across Batch: Visual tokens formed across batch for each organ.

| Modality | Methods | Liver | Right Kidney | Left Kidney | Spleen |
|---|---|---|---|---|---|
| T1w | SEEM (Baseline) | 93.89 ± 8.46 | 93.70 ± 5.80 | 90.80 ± 16.80 | 92.05 ± 16.29 |
| | w/o Visual Tokens | 95.46 ± 5.1 | 94.03 ± 9.3 | 94.43 ± 9.6 | 92.54 ± 15.8 |
| | w/o Organ Match | 96.73 ± 5.19 | 97.51 ± 5.26 | 96.64 ± 7.85 | **95.48 ± 9.79** |
| | w/o Paired Aug. | 96.86 ± 2.58 | **97.59 ± 2.53** | 97.20 ± 4.40 | 94.93 ± 12.15 |
| | Visual Across Batch | **96.95 ± 5.97** | 97.43 ± 15.58 | 96.83 ± 10.23 | 94.70 ± 16.75 |
| | CrossMR | 96.83 ± 2.58 | 97.56 ± 2.62 | **97.27 ± 4.28** | 94.97 ± 11.70 |
| T2w | SEEM (Baseline) | 77.56 ± 27.75 | 89.89 ± 9.42 | 88.67 ± 16.76 | 93.19 ± 11.26 |
| | w/o Visual Tokens | 81.84 ± 23.84 | 89.72 ± 13.95 | 92.87 ± 9.88 | 92.83 ± 11.85 |
| | w/o Organ Match | **87.81 ± 12.88** | 93.05 ± 31.13 | 90.24 ± 19.96 | 93.45 ± 7.54 |
| | w/o Paired Aug. | 85.96 ± 19.72 | 92.73 ± 8.46 | 94.68 ± 4.66 | 93.89 ± 7.87 |
| | Visual Across Batch | 84.87 ± 19.91 | 92.56 ± 10.05 | 93.39 ± 10.40 | **94.05 ± 7.70** |
| | CrossMR | 85.43 ± 2.42 | **93.96 ± 2.65** | **94.83 ± 6.63** | 93.79 ± 12.48 |
| DWI | SEEM (Baseline) | 83.88 ± 15.22 | 80.53 ± 22.84 | 82.16 ± 21.42 | 93.00 ± 11.79 |
| | w/o Visual Tokens | 88.52 ± 8.85 | 83.68 ± 19.3 | 88.30 ± 11.91 | 93.29 ± 11.90 |
| | w/o Organ Match | 88.24 ± 24.54 | 86.07 ± 14.46 | 87.87 ± 26.77 | 94.75 ± 4.29 |
| | w/o Paired Aug. | 89.22 ± 10.42 | 83.47 ± 22.74 | 87.76 ± 14.88 | 94.95 ± 3.65 |
| | Visual Across Batch | 88.55 ± 10.12 | 85.31 ± 18.97 | 89.81 ± 10.62 | **95.02 ± 3.68** |
| | CrossMR | **89.37 ± 9.86** | **88.99 ± 6.26** | **89.98 ± 3.60** | 94.81 ± 8.58 |
| In-phase | SEEM | 74.80 ± 27.08 | 66.30 ± 30.52 | 74.95 ± 23.24 | 68.51 ± 31.00 |
| | w/o Visual Tokens | 80.87 ±23.42 | 76.71 ± 23.70 | 82.91 ± 15.54 | 77.19 ± 27.96 |
| | w/o Organ Match | 83.28 ± 25.80 | 73.86 ± 31.23 | 79.94 ± 27.13 | 82.42 ± 35.98 |
| | w/o Paired Aug. | **85.83 ± 14.72** | 70.89 ± 29.99 | 77.34 ± 23.42 | 82.85 ± 21.61 |
| | Visual Across Batch | 80.73 ± 22.33 | 71.33 ± 30.80 | 79.08 ± 19.67 | 77.74 ± 25.55 |
| | CrossMR | 84.17 ±15.15 | **80.28 ± 21.73** | **83.88 ± 16.00** | **86.65 ± 16.78** |
| Opposed-phase | SEEM (Baseline) | 90.98 ± 12.61 | 86.34 ± 12.50 | 84.70 ± 16.96 | 90.24 ± 16.59 |
| | No Visual Tokens | 93.54 ± 9.05 | 86.53 ±14.10 | 88.48 ± 9.86 | 91.40 ± 14.50 |
| | w/o Organ Match | 94.38 ± 32.56 | 90.35 ± 18.14 | **89.25 ± 33.57** | 91.95 ± 18.19 |
| | w/o Paired Aug. | 94.26 ± 5.41 | 87.26 ± 16.18 | 87.46 ± 18.22 | **92.69 ± 11.56** |
| | Visual Across Batch | 93.93 ± 7.28 | 87.59 ± 17.05 | 88.37 ± 15.31 | 92.36 ± 12.16 |
| | CrossMR | **94.47 ± 5.14** | **90.68 ±7.92** | 89.11 ± 9.91 | 91.68 ± 11.67 |

computational efficiency, a maximum of 512 visual tokens across the batch are randomly chosen for self-attention, consistent with the original version. Lastly, we evaluated the performance of SEEM to observe how the overall change in the architecture impacts the performance.

Table 2 shows DSC of the ablation study results. The variants of our model consistently outperform the baseline SEEM, demonstrating CrossMR has improved performance in cross-modality referring segmentation tasks. Removing the visual tokens results in lower scores, indicating that local features from the scribble on the reference image contribute significantly to improving the performance.

CrossMR demonstrates higher overall performance compared to models without organ-specific matching, notably in in-phase modality. CrossMR also achieves a higher average DSC across all OOD modalities compared to the model without paired augmentation, suggesting that paired augmentation helps the model better integrate information from diverse contrasts between paired inputs. Lastly, the model variant using visual tokens sampled across the batch underperforms compared to CrossMR, indicating that CrossMR more effectively handles reference-target pairs trained with two augmented images.

## 5 DISCUSSION AND CONCLUSION

Motivated by the clinical demands for multi-modality MR image analysis, we have introduced the image-based referring segmentation task, which can reduce the annotation burden for the organs of interest in each MR modality. In particular, we aim to build a versatile segmentation model that can learn general representation from single-labeled reference modality and multiple unlabeled target modalities by only using scribble prompts on the reference image.

Furthermore, we have annotated a dedicated dataset for this task. We provide precise organ annotations for the reference T1w modality and four target modalities. Furthermore, we propose CrossMR, a model designed to utilize the reference image features to generate segmentation masks for various target MR modalities. CrossMR not only shows competitive performance on the in-distribution modality compared to the specialist model but also exhibits remarkable performance gains across all four out-of-distribution modalities.

CrossMR has great potential to accelerate the muti-modality MR image analysis by reducing both the cost and labor involved in the organ annotation process. By minimizing the need for extensive domain-specific knowledge, it allows segmentation across multiple MR modalities with only a single weak annotation from the reference image. Unlike models such as MedSAM-Scribble and nnUNet-Scribble, which require direct scribble input on each target image, CrossMR surpasses their performance without the need for direct annotations on every image. Scribble inputs, commonly used to guide models with positional and feature-specific information, are reduced to a single annotation on the reference modality in CrossMR, greatly easing the annotation burden.

Traditional unsupervised cross-modality segmentation models are usually designed for one target modality. In contrast, CrossMR generalizes to all five modalities, eliminating the need to develop specialized models for each target modality. This scalability becomes increasingly advantageous as the number of modalities grows, particularly in scenarios involving more than two. Additionally, compared to few-shot models, CrossMR requires only a single reference image with weak annotation for inference, whereas few-shot models rely on multiple images with full annotations. Furthermore, CrossMR outperforms the one-shot model PerSAM-F without the need for fine-tuning during inference.

This work also has limitations. The model was designed with 2D networks that can give end-users better flexibility in selecting the target slice. However, it can be extended to 3D networks to model the inter-slice information in 3D MR scans. In addition to organs, lesion segmentation in multi-modality MR images is also an important clinical task. In the future work, we will focus on adapting the network to 3D and expanding the dataset to lesion segmentation.

In conclusion, we formulate the multi-modality MR image segmentation as an image-based referring segmentation task, where users only need to draw weak supervision on a single reference image and the model can generalize to multiple target modalities. The dataset we provide could serve as a useful benchmark for this task. Moreover, it is also valuable for benchmarking more challenging tasks, where the target images are from unseen patients. In addition, CrossMR outperforms existing one-shot and scribble-based segmentation models on the target modalities, which can serve as a strong baseline model to pave the way for image-based reference segmentation.

REPRODUCIBILITY STATEMENT

To ensure the reproducibility of this work, we will make all relevant code, data, and model weights publicly available upon publication. Our code is self-contained and includes all necessary components to reproduce the results reported in the paper. Comprehensive instructions for running the experiments will be provided in the accompanying documentation. We will release the data preprocessing, training, inference, and demo code that features an interactive segmentation interface using the Gradio API. The default configuration used for the model training will also be shared. Furthermore, we will make available the fully annotated dataset consisting of five MR modalities, enabling independent validation of our results.

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

# A APPENDIX

Table 3: **Quantitative segmentation results.** Organ-wise NSD scores of the baseline model and proposed method across one in-distribution modality (T1w) and four out-of-distribution modalities (T2w, DWI, In-phase, and Opposed-phase).

| Modality | Methods | Liver | Right Kidney | Left Kidney | Spleen |
|---|---|---|---|---|---|
| T1w | nnUNe Isensee et al. (2021) | 98.87 ± 6.73 | **99.69 ± 2.04** | 98.93 ± 1.66 | 97.07 ± 12.39 |
| | PerSAM-F Zhang et al. (2023) | 79.34 ± 21.56 | 96.20 ± 15.58 | 96.57 ± 14.49 | 87.82 ± 25.57 |
| | nnUNet-Scribble Stock et al. (2024) | 98.60 ±9.64 | 99.18± 6.52 | 98.67 ±7.92 | **98.08 ±10.34** |
| | MedSAMScribble Marinov et al. (2024) | 95.62 ± 14.27 | 96.89 ± 14.84 | 94.57 ± 18.74 | 94.77 ± 15.77 |
| | Ours | **98.89 ± 1.96** | 99.61 ± 1.46 | **99.45 ± 3.09** | 97.31 ± 11.91 |
| T2w | nnUNet Isensee et al. (2021) | 71.26± 36.32 | 79.13 ±39.50 | 82.88 ± 38.66 | 87.96 ±27.95 |
| | PerSAM-F Zhang et al. (2023) | 59.05 ± 32.97 | 79.32 ± 38.73 | 69.55 ± 43.63 | 79.26 ± 35.48 |
| | nnUNet-Scribble Stock et al. (2024) | 77.98 ±11.45 | 98.64± 34.16 | 97.98±4.38 | **98.84±13.46** |
| | MedSAMScribble Marinov et al. (2024) | 78.27 ± 28.26 | 91.43 ± 22.41 | 95.80 ± 15.84 | 93.08 ± 17.62 |
| | Ours | **89.10 ± 20.63** | **98.82 ± 4.83** | **99.37 ± 1.38** | 97.28 ± 7.88 |
| DWI | nnUNet Isensee et al. (2021) | 73.81 ± 46.03 | 68.65 ± 29.75 | 63.02 ± 44.14 | 97.67 ± 8.73 |
| | PerSAM-F Zhang et al. (2023) | 55.51 ± 27.38 | 63.45 ± 46.63 | 57.48 ± 47.79 | 62.59 ± 46.44 |
| | nnUNet-Scribble Stock et al. (2024) | 85.42±21.12 | 90.38±21.88 | 91.52±24.79 | 96.90±14.42 |
| | MedSAMScribble Marinov et al. (2024) | 84.75 ± 20.17 | 88.22 ± 23.98 | 91.90 ± 19.19 | 94.75 ± 15.93 |
| | Ours | **93.60 ± 8.92** | **95.70 ± 13.58** | **96.49 ± 6.78** | **98.89 ± 2.69** |
| In-phase | nnUNet Isensee et al. (2021) | 49.50 ± 28.20 | 8.54 ± 39.86 | 13.26 ± 24.43 | 22.33 ± 35.54 |
| | PerSAM-F Zhang et al. (2023) | 50.28 ± 31.08 | 35.56 ± 45.80 | 40.48 ± 47.24 | 52.36 ± 39.88 |
| | nnUNet-Scribble Stock et al. (2024) | 60.28± 35.66 | 55.37±36.95 | 55.64±35.01 | 41.23±40.75 |
| | MedSAMScribble Marinov et al. (2024) | 73.51 ± 27.85 | 65.25 ± 33.36 | 76.37 ± 26.48 | 75.85 ± 25.43 |
| | Ours | **89.49 ±15.48** | **88.53 ± 22.47** | **92.56 ± 16.12** | **91.93 ± 17.46** |
| Opposed-phase | nnUNet Isensee et al. (2021) | 77.29 ± 38.37 | 61.84 ± 35.56 | 74.94 ± 46.20 | 77.39 ± 36.0 |
| | PerSAM-F Zhang et al. (2023) | 63.60 ± 32.23 | 58.15 ± 47.08 | 67.23 ± 44.62 | 68.72 ± 39.30 |
| | nnUNet-Scribble Stock et al. (2024) | 89.32± 17.71 | 93.52± 23.11 | 94.18 ±16.77 | 90.97± 22.03 |
| | MedSAMScribble Marinov et al. (2024) | 91.69 ± 15.29 | 88.91 ± 21.89 | 88.84 ± 22.21 | 89.73 ± 19.00 |
| | Ours | **97.03 ± 4.46** | **96.77 ±5.09** | **96.84 ± 6.24** | **95.84 ± 11.52** |

Table 4: **NSD of Ablation Study.** The results of CrossMR models with different experimental settings. Fine-tuned: trained from the SEEM pre-trained model, Encoder fixed: trained from the SEEM pre-trained model where the image encoder parameters are fixed, From scratch: trained from scratch.

| Modality | Methods | Liver | Right Kidney | Left Kidney | Spleen |
|---|---|---|---|---|---|
| T1w | SEEM (Baseline) | $97.58 \pm 8.16$ | $99.10 \pm 5.78$ | $95.58 \pm 17.37$ | $95.60 \pm 16.52$ |
| | w/o Visual Tokens | $98.52 \pm 4.94$ | $98.65 \pm 9.66$ | $98.48 \pm 9.65$ | $95.95 \pm 16.31$ |
| | w/o Organ Match | $98.81 \pm 1.97$ | $99.57 \pm 1.63$ | $98.86 \pm 8.08$ | $97.05 \pm 10.25$ |
| | w/o Paired Aug. | $98.88 \pm 1.97$ | $\mathbf{99.62 \pm 1.41}$ | $\underline{99.42 \pm 3.19}$ | $\mathbf{97.34 \pm 11.56}$ |
| | Visual Across Batch | $\mathbf{99.01 \pm 1.79}$ | $99.58 \pm 1.59$ | $99.19 \pm 6.03$ | $96.96 \pm 13.03$ |
| | CrossMR | $\underline{98.89 \pm 1.96}$ | $\underline{99.61 \pm 1.46}$ | $\mathbf{99.45 \pm 3.09}$ | $\underline{97.31 \pm 11.91}$ |
| T2w | SEEM | $81.33 \pm 28.75$ | $97.06 \pm 10.84$ | $94.97 \pm 17.24$ | $96.97 \pm 11.04$ |
| | w/o Visual Tokens | $85.81 \pm 24.74$ | $96.51 \pm 14.68$ | $98.29 \pm 9.98$ | $96.61 \pm 11.70$ |
| | w/o Organ Match | $\mathbf{91.59 \pm 16.18}$ | $\underline{98.10 \pm 7.39}$ | $94.47 \pm 21.74$ | $97.35 \pm 7.69$ |
| | w/o Paired Aug. | $\underline{89.91 \pm 19.03}$ | $98.05 \pm 6.15$ | $\underline{99.26 \pm 2.20}$ | $\underline{97.40 \pm 7.10}$ |
| | Visual Across Batch | $88.99 \pm 19.22$ | $97.73 \pm 10.91$ | $98.06 \pm 10.24$ | $\mathbf{97.64 \pm 6.98}$ |
| | CrossMR | $89.10 \pm 20.63$ | $\mathbf{98.82 \pm 4.83}$ | $\mathbf{99.37 \pm 1.38}$ | $97.28 \pm 7.88$ |
| DWI | SEEM | $88.96 \pm 14.61$ | $89.46 \pm 23.32$ | $90.27 \pm 22.29$ | $97.45 \pm 12.11$ |
| | w/o Visual Tokens | $93.51 \pm 7.65$ | $92.67 \pm 18.93$ | $95.97 \pm 8.36$ | $97.68 \pm 12.15$ |
| | w/o Organ Match | $92.59 \pm 8.93$ | $92.19 \pm 22.11$ | $94.72 \pm 13.72$ | $98.51 \pm 2.48$ |
| | w/o Paired Aug. | $\underline{93.98 \pm 9.38}$ | $90.86 \pm 22.80$ | $95.05 \pm 11.55$ | $99.03 \pm 2.49$ |
| | Visual Across Batch | $92.93 \pm 8.57$ | $\underline{92.98 \pm 17.22}$ | $\underline{96.56 \pm 7.30}$ | $\mathbf{99.04 \pm 2.47}$ |
| | CrossMR | $\mathbf{93.60 \pm 8.92}$ | $\mathbf{95.70 \pm 13.58}$ | $\mathbf{96.49 \pm 6.78}$ | $98.89 \pm 2.69$ |
| In-phase | SEEM | $79.27 \pm 28.09$ | $75.10 \pm 33.83$ | $84.83 \pm 24.87$ | $\underline{73.88 \pm 33.68}$ |
| | w/o Visual Tokens | $85.34 \pm 24.51$ | $\underline{86.08 \pm 24.74}$ | $92.51 \pm 15.95$ | $82.35 \pm 30.10$ |
| | w/o Organ Match | $87.40 \pm 19.92$ | $81.99 \pm 29.38$ | $89.18 \pm 20.87$ | $87.60 \pm 24.36$ |
| | w/o Paired Aug. | $\mathbf{90.04 \pm 14.58}$ | $79.10 \pm 32.16$ | $86.33 \pm 24.14$ | $\underline{88.39 \pm 21.60}$ |
| | Visual Across Batch | $84.50 \pm 24.10$ | $78.99 \pm 33.58$ | $88.78 \pm 19.86$ | $83.32 \pm 26.65$ |
| | Fine-tuned | $\underline{89.49 \pm 15.48}$ | $\mathbf{88.53 \pm 22.47}$ | $\mathbf{92.56 \pm 16.12}$ | $\mathbf{91.93 \pm 17.46}$ |
| Opposed-phase | SEEM | $94.92 \pm 12.79$ | $94.59 \pm 12.72$ | $92.94 \pm 16.84$ | $93.94 \pm 16.77$ |
| | No Visual Tokens | $96.75 \pm 8.78$ | $94.38 \pm 13.47$ | $\underline{96.22 \pm 7.25}$ | $95.05 \pm 15.12$ |
| | w/o Organ Match | $\mathbf{97.05 \pm 4.57}$ | $\underline{96.73 \pm 6.35}$ | $95.20 \pm 13.98$ | $95.33 \pm 12.73$ |
| | w/o Paired Aug. | $97.03 \pm 4.43$ | $94.04 \pm 13.75$ | $93.79 \pm 17.22$ | $\mathbf{96.01 \pm 11.07}$ |
| | Visual Across Batch | $96.75 \pm 7.17$ | $93.53 \pm 17.07$ | $94.83 \pm 13.09$ | $95.41 \pm 12.82$ |
| | CrossMR | $97.03 \pm 4.46$ | $\mathbf{96.77 \pm 5.09}$ | $\mathbf{96.84 \pm 6.24}$ | $95.84 \pm 11.52$ |