# OpenReview forum: "All You Need Is A Reference: Cross-modality Referring Segmentation for Abdominal MRI"
_ICLR.cc/2025/Conference — Submitted to ICLR 2025_

### Official Review · Reviewer_nPdD · 2024-10-31

**Soundness:** 2
**Presentation:** 2
**Contribution:** 2
**Rating:** 1
**Confidence:** 5

**Summary:**

In this paper, the authors introduced an image-based referring segmentation approach that takes from users a simple scribble on a T1w image and outputs the segmentation of multiple other modalities, including T2w, DWI, In-phase, and Opposed-phase MRI, in multi-modality MRI scans.

**Strengths:**

This work introduces the first referring segmentation task for organ segmentation in multi-modality MRI.

**Weaknesses:**

1. While the method appears novel, it seems the authors are tackling a relatively simple problem in a convoluted manner. The issue could be effectively addressed by training a segmentation model for T1w images and using a rigid registration method to transfer labels from T1w images to other modalities.
2. The workings of paired data augmentation are unclear. Translating a T1w image into an image resembling T2w, DWI, In-phase, and Opposed-phase MRI is not straightforward, especially at high resolutions. Are there any existing methods that can perform this task accurately and reliably? How does they work?
3. There is a lack of an overview of the proposed approach, making it unclear what role each subsection in Section 3 plays.

**Questions:**

1. Could you clarify the motivation behind your method? It seems to me that the target problem could be effectively addressed by training a multi-organ segmentation model on T1w images and subsequently using a rigid registration method to transfer labels from T1w images to other modalities. Rigid registration is very reliable for intra-subject alignment. And I think this simpler method can be more advantageous as it doesn't require any inputs from users during inference. If I have misunderstood the problem, please explain why a simpler approach like this would not be suitable. If my understanding is correct, could you elaborate on the advantages of your proposed method?
2. Please explain the rationale for first interacting the spatial queries $Q$ with the target image features through cross-attention, followed by concatenating the visual tokens for each organ with the organ’s spatial queries, and finally applying self-attention within the combined queries. What does each step accomplish, and why is this specific order important?
3. How many object queries are used for each organ? Why this number is adopted? And why the object queries are duplicated for each organ?
4. Could you provide a detailed explanation of how paired data augmentation works? Specifically, how do you translate a T1w image into an image resembling T2w, DWI, In-phase, and Opposed-phase MRI?

---

### Official Review · Reviewer_gjxJ · 2024-11-02

**Soundness:** 2
**Presentation:** 2
**Contribution:** 3
**Rating:** 5
**Confidence:** 4

**Summary:**

In this work, the authors introduce a novel cross-modality referring segmentation task, where users can guide multi-modality segmentation by simply drawing scribbles on one reference modality, significantly reducing annotation costs. The proposed method demonstrates competitive results on their created MRI dataset.

**Strengths:**

1. The approach requires only one annotation to achieve multi-modality segmentation, avoiding individual annotations for each target image and simplifying the workflow.
2. The authors constructed a multi-modality MRI dataset with five MRI modalities and four abdominal organs, which can advance research in cross-modality segmentation.

**Weaknesses:**

1. Despite the authors mentioning that they do not focus on multi-stage cross-modality approaches, the introduction and analysis are overly simplified. The experimental section lacks performance and computational complexity comparisons with cross-modality methods. Over-emphasizing the superiority of the method under its own setup makes the results less convincing. It is recommended to include relevant multi-stage cross-modality methods and conduct comparative analysis of computational speed and parameter counts between methods.
2. In the experimental setup, the authors only consider the scenario where T1w is used as the reference modality. This raises questions about whether the method is effective in other scenarios. More reference modality settings need to be considered to validate the generalizability of the method.
3. While contributing a new dataset is commendable, conducting experiments solely on a private dataset makes it difficult to obtain a fair and convincing evaluation of effectiveness. It is suggested to perform comparisons using other public datasets such as BraTS, as well as on lesion segmentation tasks.

Minors: The introduction of Table 4 seems to be mismatched with its content.

**Questions:**

All the concerns are mentioned in the weaknesses. It should be emphasized that the authors should focus on increasing the diversity of experimental settings to validate the effectiveness of their method.

---

### Official Review · Reviewer_twfB · 2024-11-03

**Soundness:** 2
**Presentation:** 3
**Contribution:** 2
**Rating:** 3
**Confidence:** 5

**Summary:**

The author introduces CrossMR, a method for cross-modality referring segmentation in multi-modal abdominal MRI. The authors propose a model that leverages weak annotations (scribbles) on a reference modality (T1w MRI) to guide the segmentation of target modalities (T2w, DWI, In-phase, and Opposed-phase MRI). A new multi-modal abdominal MRI dataset is also introduced, and the method is evaluated on segmenting four labeled organs across multiple modalities.

**Strengths:**

1. The paper is clearly written and logically structured, making the technical contributions easy to follow.

2. The paper introduces a new multi-modal abdominal MRI dataset with annotations for four organs, which is a useful contribution to the medical imaging community.

**Weaknesses:**

1. A major concern with the task definition is the unclear motivation for referring segmentation when target images are well co-registered with the reference images. In a co-registered setting, where both the reference and target modalities share the same ground truth, the output masks should be identical for both modalities. If co-registration is strong, a straightforward solution would be to perform segmentation directly on the reference image (e.g., T1w) and apply the same output mask to the target images. In this case, as shown in the results (Table 1), nnUNet would be the preferred model based on its performance on the T1w reference modality. This raises the question: Why is referencing segmentation necessary in such a scenario, and how does it add value over simpler approaches like direct segmentation on the reference modality?

2. The paper lacks significant methodological novelty. The approach essentially adapts existing natural image segmentation techniques (i.e., referring segmentation) to the medical imaging domain without introducing substantial innovations. The proposed method relies heavily on a segmentation framework designed for natural images, such as SEEM, with only minor adjustments for multi-modal MRI data. There is no substantial architectural change or novel mechanism introduced for the medical imaging context.

3. The model is evaluated on a single dataset that contains only four labeled organs. This is a significant limitation, as it restricts the generalizability of the results to other medical imaging datasets or tasks. To demonstrate the broader applicability of CrossMR, the authors should test the method on other datasets or scenarios with more organs or different imaging modalities (e.g., CT or x-ray).

**Questions:**

1. How does CrossMR handle the case where the reference modality (T1w) is unavailable? What would the segmentation strategy be in such cases?

2. What would happen if CrossMR were trained using multi-modal input data without referencing? Would it improve performance in both in-distribution and out-of-distribution settings?

---

### Official Review · Reviewer_HdZA · 2024-11-04

**Soundness:** 3
**Presentation:** 2
**Contribution:** 2
**Rating:** 6
**Confidence:** 4

**Summary:**

The paper introduces a framework for cross-modality abdominal MRI segmentation by referring to modality. Specifically, the authors introduce a new dataset with annotations across five MRI modalities, and present a model, CrossMR, to utilize simple scribbles on a single reference modality to generate segmentations on other modalities. Experimental results on both in-distribution (ID) and out-of-distribution (OOD) modalities show promising performance. This approach can help reduce annotation burdens and enhance cross-modality segmentation applications in clinical settings.

**Strengths:**

1. Novel task design: the referring segmentation task is a creative solution to the challenge of cross-modality segmentation.
2. Dataset contribution: the new dataset, comprising 534 cases and 3,277 organ masks across five MRI modalities, could serve as a foundation for future research on multi-modality MRI segmentation.
3. Method: the proposed solution, CrossMR, which utilizes paired data augmentation and organ-specific bipartite matching, is reasonable.
4. Experiments: the authors rigorously benchmark CrossMR against state-of-the-art models and show its effectiveness.

**Weaknesses:**

1. Motivation of the method: motivation and details about the proposed method should be illustrated more clearly.
2. Weakness of ablations: the results are not well illustrated, and the effectiveness of augmentation is doubtful.
3. 2D model: the method is purely based on 2D slices. Experts still need to provide a scribble for each organ in every slice.

**Questions:**

1. What is the major contribution in terms of the referring segmentation? A related line of research is that a model is only trained with image-scribble pairs and a prototype matching method is utilized. What are the advantages of CrossMR?

2. Paired data augmentation
2.1 How are they chosen?
2.2 What advantages do they bring?
2.3 According to the ablation study, augmentation does not seem very useful.

3. It is beneficial to show some qualitative results of ablation studies. For example, if bipartite matching is not used, what results will be obtained?

---

### Official Review · Reviewer_PE2e · 2024-11-04

**Soundness:** 2
**Presentation:** 3
**Contribution:** 3
**Rating:** 5
**Confidence:** 4

**Summary:**

This work addresses the effective segmentation of abdominal organs in multi-modality MRI. Using basic scribbles annotations on a single reference MRI modality (T1-weighted), the authors provide a novel referencing segmentation task to direct segmentation across several unexplored target modalities (T2-weighted, DWI, In-phase, and Opposed-phase MRI). This work is benchmarked using a fresh dataset including 3,277 segmented organs spread over five MRI modalities.  While showing improved generalizing ability on out-of-distribution (OOD) modalities, experimental data show that the proposed method performs comparatively to leading segmentation models on the in-distribution (ID) modality (T1-weighted).

**Strengths:**

1.The authors create and annotate a substantial multi-modality MRI dataset, covering 534 cases with 3,277 segmentation masks across five modalities and four abdominal organs.

2.The proposed CrossMR model shows the ability to segment several target modalities with only scribbling annotations on a single reference modality, potentially decreasing clinician workload.

3.The method demonstrates good results when applied to out-of-distribution target modalities, outperforming existing approaches.

**Weaknesses:**

1.While the paper mentions automatic scribble generation, it doesn't provide sufficient information on how well this mimics real clinical annotations or its impact on model performance.

2.While the writing is generally good, some technical aspects would benefit from further details. For instance, section 3.4 (Automatic Scribble Generation) could provide more details on how different scribble patterns impact segmentation accuracy.

3.More study of the performance heterogeneity between modalities (such as opposed-phase vs. in-phase) will be better. Investigating if modality-specific factors like as contrast variations affect segmentation accuracy could provide more insight into the model's robustness and limitations.

**Questions:**

1. What are the computational demands for CrossMR training and inference? How does this compare to current methods?
2.Could you detail on the automatic scribble generating method and how accurately it appears like real clinical annotations?
3.How robust is the method to variations in image quality or artifacts that are common in clinical MRI scans?

---

### Meta-Review · Area_Chair_tyhw · 2024-12-19

**Metareview:**

The paper introduces an innovative referring segmentation task, a new multi-modality MRI dataset, and a method (CrossMR) that reduces annotation burdens while demonstrating promising results on target modalities. However, the paper lacks sufficient methodological novelty, clarity in motivation, and detailed ablation studies. The use of 2D slices and reliance on annotations for each slice limit scalability, while the experimental scope is restricted to a single dataset. Three reviewers rated the paper marginally above/below the acceptance threshold (5 and 6), one reviewer rejected it (3) and one reviewer strongly rejected it (1).

Besides, no rebuttal was provided and all reviewers suggested "reject" this paper. So I recommend "reject".

**Additional Comments On Reviewer Discussion:**

No rebuttal.

---

### Decision · Program_Chairs · 2025-01-22

Reject